# TOWARDS BUILDING RELIABLE CONDITIONAL DIFFUSION MODELS FOR PROTEIN GENERATION

## ABSTRACT

Generating novel and functional protein sequences is critical to a wide range of applications in biology. Recent advancements in conditional diffusion models have shown impressive empirical performance in protein generation tasks. However, reliable generations of protein remain an open research question in *de novo* protein design, especially when it comes to conditional diffusion models. Considering the biological function of a protein is determined by multi-level structures, we propose a novel multi-level conditional diffusion model that integrates both sequence-based and structure-based information for efficient end-to-end protein design guided by specified functions. By generating representations at different levels simultaneously, our framework can effectively model the inherent hierarchical relations between different levels, resulting in an *informative and discriminative representation* of the generated protein. We also propose a Protein-MMD, a new reliable evaluation metric, to evaluate the quality of generated protein with conditional diffusion models. Our new metric is able to capture both distributional and functional similarities between real and generated protein sequences while ensuring conditional consistency. We experiment with standard datasets and the results on protein generation tasks demonstrate the efficacy of the proposed generation framework and evaluation metric.

## 1 INTRODUCTION

Designing proteins with specific biological functions is a fundamental yet formidable challenge in biotechnology. It benefits wide-ranging applications from synthetic biology to drug discovery (Watson et al., 2023; Bose et al., 2016; Huang et al., 2016; Feng et al., 2024; Huang et al., 2024b;b; Lin et al., 2023). The challenge arises from the intricate interplay between protein sequence, structure, and function, which has not yet been fully understood (Dill et al., 2008). Traditional methods, such as directed evolution, rely on labor-intensive trial-and-error approaches involving random mutations and selective pressures, making the process time-consuming and costly (Arnold, 1998). Recently, generative models have emerged as promising tools for protein design, enabling the exploration of vast sequence-structure-function landscapes (Anand & Achim, 2022; Fu et al., 2024; Dauparas et al., 2022; Trippe et al., 2023). However, existing generative models—including those focused on enzyme engineering, antibody creation, and therapeutic protein development—are typically task-specific and require retraining for new design objectives (Fu et al., 2024; Dauparas et al., 2022). These limitations impede their adaptability and scalability across different protein families.

While conditional generative models offer an end-to-end solution by directly linking the design process to the guidance, these models have been applied to protein generation Komorowska et al. (2024); Klarner et al. (2024); Gruver et al. (2023). In conditional protein generation tasks, maintaining conditional consistency across diverse contexts and ensuring functional relevance are critical (Trippe et al., 2022; Hu et al., 2024). Specifically, the generated proteins should fully adhere to the specified functional constraints (Gretton et al., 2012). At the same time, achieving diversity and novelty in generated proteins is essential for successful design. In the literature, structural novelty can be assessed using Foldseek (van Kempen et al., 2022), which performs rapid protein structure searches against databases like PDB (Berman et al., 2000) and AlphaFold (Jumper et al., 2021) to ensure the generated proteins are novel compared to known structures. Diversity is measured using TM-score (Zhang & Skolnick, 2005), which calculates structural variation between the generated proteins themselves and between the generated and wild-type proteins (Hu et al., 2024).

However, it is unknown how to assess the *conditional consistency* (Gretton et al., 2012) in *de novo* protein design. Specifically, the fundamental problem of properly evaluating conditional consistency is quantifying to what extent the generated protein adheres to the specified functional constraints. Unlike computer vision, where metrics such as FID (Heusel et al., 2017) have become a standard for assessing generated images, it is unclear whether such metrics are suitable for protein generation tasks. In protein design, the generated output cannot be as easily visualized or assessed as in images, making the choice of evaluation metrics even more critical. Therefore, how to adapt metrics like FID or Maximum Mean Discrepancy (Gretton et al., 2012) presents challenges. Furthermore, existing methods for protein generation either generate the representation at a single level or ignore hierarchical relations within different levels of protein representation (e.g., amino acid and atom). Choosing the level of granularity at which capturing both the structural and functional nuances of protein sequences introduces additional challenges, raising significant concerns about the reliability of generated proteins in real-world applications. This difficulty highlights the need for a more rational generative scheme, making the generated protein adhere to the specified functional constraints.

Motivated by the need to capture both the structural and functional nuances of protein sequences, we propose a novel multi-level conditional generative diffusion model for protein design that integrates both sequence-based (Lin et al., 2023) and structure-based (Wang et al., 2022a) hierarchical information. Generation at multi-levels enables efficient end-to-end generation of proteins with specified functions and modeling the inherent hierarchical relations between different representations, resulting in an informative and discriminative representation of the protein. Our model incorporates a rigid-body 3D rotation-invariant preprocessing step combined with an autoregressive decoder to maintain SE(3)-invariance, ensuring accurate modeling of protein structures in 3D space. To address the challenges of evaluating the *conditional consistency*, we propose *Protein-MMD*, a metric based on Maximum Mean Discrepancy (MMD), to better capture both distributional and functional similarities between real and generated protein sequences, while ensuring conditional consistency. We prove that our Protein-MMD provides a more accurate measure that reflects the given condition. Experiments demonstrate that our proposed model outperforms existing approaches in generating diverse, novel, and functionally relevant proteins.

Our main contributions are summarized as follows:

- We design a novel multi-level conditional generative diffusion model that integrates sequence-based and structure-based information for efficient end-to-end protein design. Our model incorporates a rigid-body 3D rotation-invariant preprocessing step to maintain SE(3)-invariance, ensuring accurate modeling of protein structures in 3D space.

- We highlight the limitations of current evaluation metrics in protein generation tasks, particularly in conditional settings, and propose *Protein-MMD*, a novel metric to evaluate conditional consistency for protein generation tasks by leveraging language models.

- We experiment with standard datasets to verify the effectiveness of the proposed model. Our evaluation metric paves the way for reliable protein design with given conditions.

## 2 PRELIMINARY

### 2.1 DIFFUSION MODELS

Denoising Diffusion Probabilistic Models (DDPMs) (Ho et al., 2020) are a class of generative models that learn to model data distributions by iteratively denoising data corrupted with Gaussian noise. The forward diffusion process gradually adds noise to the data, transforming the complex data distribution into a tractable Gaussian distribution:

$$q(\mathbf{x}_t \mid \mathbf{x}_{t-1}) = \mathcal{N}(\mathbf{x}_t; \sqrt{1 - \beta_t}\mathbf{x}_{t-1}, \beta_t \mathbf{I}), \tag{1}$$

where $\mathbf{x}_0$ is the original data, $\mathbf{x}_t$ is the data at timestep $t$, $\beta_t$ is the variance schedule, and $\mathcal{N}$ denotes a Gaussian distribution. The reverse diffusion process aims to model the posterior $q(\mathbf{x}_{t-1} \mid \mathbf{x}_t)$, which is approximated by a neural network $\epsilon_\theta$ parameterized by $\theta$:

$$p_\theta(\mathbf{x}_{t-1} \mid \mathbf{x}_t) = \mathcal{N}(\mathbf{x}_{t-1}; \mu_\theta(\mathbf{x}_t, t), \Sigma_\theta(\mathbf{x}_t, t)), \tag{2}$$

where $\mu_\theta$ and $\Sigma_\theta$ are functions of the current data $\mathbf{x}_t$ and timestep $t$.

The model is trained to minimize the variational bound on negative log-likelihood, which simplifies to a mean squared error loss between the predicted noise and the true noise:

$$L_{\text{simple}} = \mathbb{E}_{\mathbf{x}_0, \epsilon, t}\left[\left|\epsilon - \epsilon_\theta(\mathbf{x}_t, t)\right|^2\right], \tag{3}$$

where $t$ is uniformly sampled from $1, \ldots, T$. Instead of using traditional U-Net architectures, recent works Peebles & Xie (2023a) leverage Transformer to predict the noise in the latent space (Rombach et al., 2022):

$$\epsilon_\theta(\mathbf{z}_t, t) = \text{Transformer}(\mathbf{z}_t, t; \theta), \tag{4}$$

where $\mathbf{z}$ is a latent representation of $\mathbf{x}$ and $\theta$ includes the parameters of the Transformer model.

## 2.2 FRÉCHET INCEPTION DISTANCE

Fréchet Inception Distance (FID) is commonly used in computer vision to measure the similarity between real and generated data distributions (Heusel et al., 2017). It computes the Fréchet distance between feature representations of real and generated samples, extracted from a pre-trained network, such as Inception v3 (Szegedy et al., 2016). The FID between two distributions $p_r$ (real data) and $p_g$ (generated data) is computed by modeling the features as multivariate Gaussian distributions with means $\mu_r, \mu_g$ and covariances $\Sigma_r, \Sigma_g$:

$$\text{FID}(p_r, p_g) = \|\mu_r - \mu_g\|_2^2 + \text{Tr}\left(\Sigma_r + \Sigma_g - 2(\Sigma_r \Sigma_g)^{1/2}\right), \tag{5}$$

where $\|\cdot\|_2$ denotes the Euclidean distance, and Tr is the trace operator. FID provides a measure of how close the real and generated data are, with lower FID values indicating high similarity.

## 3 METHODOLOGY

### 3.1 MULTI-LEVEL DIFFUSION

Motivated by the need to capture both the structural and functional nuances of protein sequences, we propose a multi-level diffusion model to generate information about a protein at three levels: the amino acid level, the backbone level, and the all-atom level. By constructing representations at different levels, our framework effectively integrates the inherent hierarchical relations of proteins, resulting in a more rational protein generative model. We remark that there are hierarchical relations among different levels. To the best of our knowledge, this work is the first diffusion model to generate information at three levels and leverage the hierarchical relation between different levels.

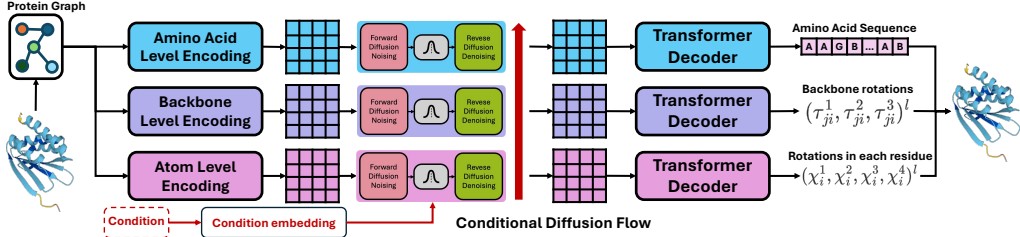

Figure 1: The architecture of the multi-level diffusion model

Figure 1 shows the architecture of our model. At each level, the information will be encoded with its own set of embeddings and processed through a conditional diffusion flow where the condition comes from a lower level. With decoders, the sequence, backbone rotations, and residue rotations will be combined to indicate the complete information of a generated protein.

**Amino Acid Level Representation:** As the 3D conformation dictates biochemical interactions (Huang et al., 2016; Dill et al., 2008), we first represent a protein's structure as a graph $\mathcal{G}_a = (\mathcal{V}_a, \mathcal{E}_a)$, where $\mathcal{V}_a$ is the set of nodes corresponding to residues (amino acids), and $\mathcal{E}_a$ is the set of edges representing interactions between two residues if their Euclidean distance in 3D space is below a certain threshold, indicating potential interactions between them. At the amino acid level, each node $v_i \in \mathcal{V}_a$ corresponds to an amino acid and is represented by a vector $\boldsymbol{v}_i = (\boldsymbol{\phi}_i; \mathbf{h}_i)$, where $\boldsymbol{\phi}_i \in \mathbb{R}^3$ denotes the spatial coordinates of the amino acid's C$\alpha$ atom in three-dimensional space,

and $\mathbf{h}_i$ abstracts biochemical or structural properties. Each edge is represented as an embedding of the sequential distance Zhang et al. (2023).

**Backbone Level Representation:** An amino acid consists of backbone atoms and side chain atoms. Similarly, we use backbone atom ($C$, $N$, $C_\alpha$) coordinates as the feature of in node of the backbone $\mathcal{V}_b$. We follow Zhang et al. (2023) to compute three Euler angles $\tau_{i,j}^1$, $\tau_{i,j}^2$, $\tau_{i,j}^3$ between two backbone atoms $i$ and $j$. The angles will be integrated with the sequential distance as the edge feature. Backbone-level representation derives finer-grained protein information. With the three angles, the orientation between any two backbone planes can be determined to capture the backbone structures.

**Atom Level Representation:** Atom-level representation considers all atoms in the protein and provides the most fine-grained information. There are several methods to treat an atom as a node in the representation Hermosilla et al. (2021); Jing et al. (2021). Side chain torsion angles are important properties of protein structures Jumper et al. (2021). In this paper, we also consider geometric representation at the atom level by incorporating the first four torsion angles: $\chi_i^1$, $\chi_i^2$, $\chi_i^3$, and $\chi_i^4$. With the complete geometric representation at the atom level, the diffusion model can capture 3D information about all atoms in a protein and distinguish any two distinct protein structures in nature.

**Encoding:** We adopt a graph neural networks model Wang et al. (2022b) to encode the representing at different levels by leveraging the message-passing mechanism. In many models dealing with the spatial positions of amino acids, SE(3)-equivariance is often leveraged to ensure the invariance of operations such as translation and rotation Bose et al. (2016); Yim et al. (2023). We also introduce a novel method to ensure SE(3)-invariance by transforming each amino acid's coordinates $\phi$. This step is crucial for facilitating the subsequent autoregressive decoding.

Given a protein chain, we first translate the coordinates such that the position of the first amino acid is moved to the origin, i.e., $(0, 0, 0)$. Then, we apply a rotation matrix to align the position of the second amino acid onto the positive $x$-axis:

$$R_1 = I + \sin(\theta)K + (1 - \cos(\theta))K^2, \tag{6}$$

where $\theta$ is the rotation angle between a node $v$ and the $x$-axis, and $K$ is the skew-symmetric matrix derived from the cross-product of $v$ and the unit vector along the $x$-axis. The third amino acid is rotated around the $x$-axis to place it in the positive $xy$-plane:

$$\mathbf{R}_2 = \begin{pmatrix} 1 & 0 & 0 \\ 0 & \cos(\psi) & -\sin(\psi) \\ 0 & \sin(\psi) & \cos(\psi) \end{pmatrix}, \tag{7}$$

where $\psi$ is the angle that brings the third amino acid into the $xy$-plane. This process is iteratively applied to all amino acids in the protein chain.

The decoder at each level is an autoregressive Transformer (Vaswani, 2017) model that reconstructs the protein at each respective level. The autoregressive decoder can then use these transformed embeddings to reconstruct the information of a protein. At the sequence level, the decoder predicts the next amino acid token in the sequence. At the backbone level and the atom level, the decoder predicts geometric features (e.g., bond angles and distances) in an autoregressive fashion of each amino acid in the protein chain. Our method facilitates the use of SE(3)-invariant embeddings within an autoregressive framework. The decoder's autoregressive nature allows it to progressively predict amino acid positions by leveraging the SE(3)-invariant representation.

**Proof of SE(3)- invariance of the Transformation**

Let $\{\phi_i\}_{i=1}^n$ be the original coordinates of the amino acids in the protein chain. Consider an arbitrary rotation $\mathbf{R} \in \mathrm{SO}(3)$ and translation $\boldsymbol{\Gamma} \in \mathbb{R}^3$ applied to the protein, resulting in transformed coordinates:

$$\phi_i' = \mathbf{R}\phi_i + \boldsymbol{\Gamma}. \tag{8}$$

Our goal is to show that after applying the transformation method to both $\{\phi_i\}$ and $\{\phi_i'\}$, the resulting representations are identical.

*Proof*: For any transformation $T$ in $\mathrm{SO}(3)$ and any vector $\mathbf{v} \in \mathbb{R}^3$, we have:

$$T(\mathbf{v}) = \mathbf{R}\mathbf{v}. \tag{9}$$

Since rotations preserve vector norms, we can express $T(\mathbf{v})$ in terms of the norm of $\mathbf{v}$ and its unit vector $\mathbf{v}' = \mathbf{v}/\|\mathbf{v}\|$:

$$T(\mathbf{v}) = \|\mathbf{v}\|\mathbf{R}\mathbf{v}' = \|\mathbf{v}\|T(\mathbf{v}'). \tag{10}$$

This implies that the effect of $T$ on $\mathbf{v}$ can be decomposed into scaling by $\|\mathbf{v}\|$ and transforming its direction via rotation and translation. To simplify the expression and subsequent calculations, we denote all vectors $\phi_i$ as unit vectors (i.e., their norms are equal to 1).

*Step 1:* Translation to Origin Compute the relative positions with respect to the first amino acid:

$$\boldsymbol{\xi}_i = \boldsymbol{\phi}_i - \boldsymbol{\phi}_1, \tag{11}$$

$$\boldsymbol{\xi}'_i = \boldsymbol{\phi}'_i - \boldsymbol{\phi}'_1 = (\mathbf{R}\boldsymbol{\phi}_i + \boldsymbol{\Gamma}) - (\mathbf{R}\boldsymbol{\phi}_1 + \boldsymbol{\Gamma}) = \mathbf{R}(\boldsymbol{\phi}_i - \boldsymbol{\phi}_1) = \mathbf{R}\boldsymbol{\xi}_i. \tag{12}$$

Thus, we have $\boldsymbol{\xi}'_i = \mathbf{R}\boldsymbol{\xi}_i$.

*Step 2:* Rotation to Align Second Amino Acid Along Positive $x$-Axis: since $\|\boldsymbol{\xi}_2\| = \|\boldsymbol{\xi}'_2\| = 1$, we have:

$$\mathbf{R}_1\boldsymbol{\xi}_2 = \mathbf{e}_x, \tag{13}$$

$$\mathbf{R}'_1\boldsymbol{\xi}'_2 = \mathbf{e}_x, \tag{14}$$

where $\mathbf{e}_x = [1, 0, 0]^\top$. Since $\boldsymbol{\xi}'_2 = \mathbf{R}\boldsymbol{\xi}_2$, we have:

$$\mathbf{R}'_1\mathbf{R}\boldsymbol{\xi}_2 = \mathbf{e}_x. \tag{15}$$

Let $\mathbf{R}'_1 = \mathbf{R}_1\mathbf{R}^{-1}$, then:

$$\mathbf{R}'_1\boldsymbol{\phi}'_i = \mathbf{R}_1\mathbf{R}^{-1}\mathbf{R}\boldsymbol{\phi}_i = \mathbf{R}_1\boldsymbol{\phi}_i. \tag{16}$$

*Step 3:* Rotation Around $x$-Axis to Place Third Amino Acid in $xy$-Plane. Find rotation matrices $\mathbf{R}_2$ and $\mathbf{R}'_2$ (rotations around the $x$-axis) such that:

$$\mathbf{R}_2\mathbf{R}_1\boldsymbol{\phi}_3 \in \text{span}\{\mathbf{e}_x, \mathbf{e}_y\}, \tag{17}$$

$$\mathbf{R}'_2\mathbf{R}'_1\boldsymbol{\phi}'_3 \in \text{span}\{\mathbf{e}_x, \mathbf{e}_y\}. \tag{18}$$

Since $\mathbf{R}'_1\mathbf{d}'_3 = \mathbf{R}_1\mathbf{d}_3$, we have:

$$\mathbf{R}'_2\mathbf{R}_1\boldsymbol{\phi}_3 = \mathbf{R}_2\mathbf{R}_1\boldsymbol{\phi}_3. \tag{19}$$

Thus, $\mathbf{R}'_2 = \mathbf{R}_2$. After applying the sequence of transformations, the final coordinates are:

$$\tilde{\boldsymbol{\phi}}_i = \mathbf{R}_2\mathbf{R}_1\boldsymbol{\phi}_i, \tag{20}$$

$$\tilde{\boldsymbol{\phi}}'_i = \mathbf{R}'_2\mathbf{R}'_1\mathbf{d}'_i = \mathbf{R}_2\mathbf{R}_1\boldsymbol{\phi}_i = \tilde{\boldsymbol{\phi}}_i. \tag{21}$$

Thus, $\tilde{\boldsymbol{\phi}}'_i = \tilde{\boldsymbol{\phi}}_i$, proving that the transformed coordinates are invariant under any initial rotation $\mathbf{R}$ and translation $\boldsymbol{\Gamma}$. This confirms that the method achieves SE(3)-invariance.

**Hierarchical Diffusion with Conditional Flow:** To achieve control over the conditional generation of proteins at multiple levels, we employ a novel hierarchical diffusion model with a conditional flow mechanism. This design enables fine-grained manipulation of protein structure generation under specific conditions, such as targeted functional attributes. The diffusion process is split into three distinct levels: all-atom, backbone, and amino acid (sequence). Conditional information is injected from a lower level to ensure conditional consistency throughout the generation process.

Our conditional flow mechanism facilitates the transfer of information from lower levels (atom) to higher levels (backbone and amino acid) during the generation process. After denoising at each level, the latent representation is passed upward through a linear projection. Figure 2 shows the conditional flow (red lines). Specifically, for each level, the conditional flow integrates the latent

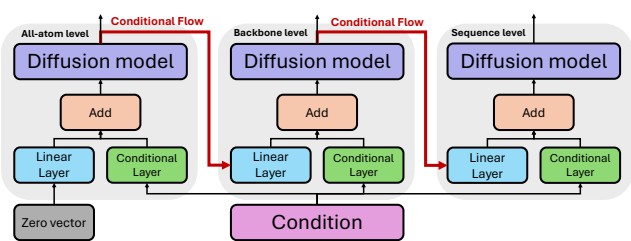

Figure 2: Hierarchical diffusion model

vector from the lower level through a projection operation, which aligns the latent vector of the lower level to the higher level's embedding space via a learned linear transformation. This ensures that the structural information from the previous level is preserved and effectively conditions the next level's generation. The input at the atom level starts as a zero vector $z_t^0 = \mathbf{0} \in \mathbb{R}^{L \times d}$. At the

higher levels, the latent vector from the previous level, after removing noise, is linearly projected and combined with the current level's conditional embedding and time step embedding to ensure that the generative process is guided by both the condition and the structural information from the lower levels. The update at level $i$ is given by:

$$z_t^i = \epsilon^i(z_{t-1}^i; z_t^{i-1} W^i, c, \gamma_t), \tag{22}$$

where $z_t^i \in \mathbb{R}^{L \times d}$ is the latent vector at level $i$ and time step $t$, $z_t^{i-1}$ is the latent vector from the previous level, $W^i \in \mathbb{R}^{d \times d}$ is a learned linear projection matrix, $c$ represents the conditional embedding (e.g., the protein's functional target), and $\gamma_t$ is the time step embedding. Denote $\epsilon^i$ as the diffusion model at level $i$, which predicts the noise added during the forward process.

---

**Algorithm 1** Training Diffusion Models with Conditional Flow

---

1: **while** epoch $<$ epochs **do**
2:  Sample a random timestep $t$
3:  **for all** levels $i \in \{1, 2, 3\}$ **in parallel do**
4:    **if** $i = 1$ **then**
5:      Initialize zero vector $z_t^0$
6:    **else**
7:      Initialize $z_0^{i-1}$ from ground truth data
8:    **end if**
9:    Sample noise vectors
10:    Diffuse latent vectors to get $z_t^{i-1}$ and $z_{t-1}^i$
11:    Update latent vector:
12:      $z_t^i \leftarrow \epsilon^i(z_{t-1}^i; z_t^{i-1} W^i, c, \gamma_t)$
13:    Compute loss at $i$th level
14:    Update model parameters
15:  **end for**
16:  epoch$+ = 1$
17: **end while**

---

**Training with Teacher Forcing:** To enable efficient parallel training, we use the teacher forcing method during training. In this setup, for each level, the input $z_0^{i-1}$ to the conditional flow is the ground truth data from the previous level, rather than the model's own generated output. This allows us to decouple the training of the three levels, enabling them to be trained independently and in parallel. The training process for the diffusion model at each level follows the typical DDPM framework but with the conditional flow incorporated to introduce additional control over the generative process. The training procedure is outlined in Algorithm 1.

## 3.2 EVALUATION OF CONDITIONAL CONSISTENCY

Evaluating the quality and consistency of protein generation models requires a well-defined framework, particularly in the context of conditional generation. In this section, we define the theoretical basis for assessing conditional consistency in multi-class generation tasks and propose a novel framework to assess the suitability of different evaluation metrics.

Denote $\{C^1, C^2, \ldots, C^K\}$ as a set of target classes, where each class $C^k$ corresponds to an independent and mutually exclusive category (e.g., different protein functions or classes). Let $\mathbf{x}$ represent a data sample, and $d(\mathbf{x}, C_k)$ is a conditional consistency metric that measures the consistency between a sample $x$ and the target class $C_k$. Given a model exhibiting strong conditional consistency, it should generate samples such that as we progress through a sequence of generated samples $\{\mathbf{x}^i | i = 0, 1, 2, \cdots, \infty\}$ ordered by increasing quality, the consistency distance between each sample and samples in $C^k$ should decrease. Mathematically, a good evaluation metric $d$ satisfies:

$$\lim_{i \to \infty} d(\mathbf{x}^i, C^k) \to 0. \tag{23}$$

It implies that as the sample quality improves, the consistency to the correct target class decreases asymptotically towards zero. We can further derive the following theorem.

**Theorem:** $\exists N \in \mathbb{N}^+, \forall i > N, d(\mathbf{x}^i, C^k) < \min_{j \neq k} d(\mathbf{x}^i, C^j)$ where $C^j$ is any other class.

*Proof*: see Appendix A.1.

Given that test samples exhibit strong conditional consistency, the theorem suggests that if we measure $d(\cdot)$ between test samples and all target classes, the majority will be classified into the correct target class $C^k$. However, relying solely on spatial distance may be too rigid for general evaluation, especially in conditional settings. In Figure 3, the green points represent generated samples, and darker shades indicate better sample quality. A well-defined metric should indicate that the green points are closer to their correct target class (i.e., Class 2) rather than the blue or pink classes.

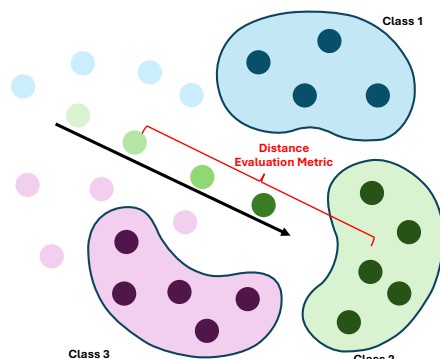

Figure 3: Consistency in the latent space

Besides the accuracy (which class the generated belongs to), Mean Reciprocal Rank (MRR) and Normalized Mean Rank (NMR) are widely used to assess how well the evaluation metric ranks generated samples based on their correct target classes. Specifically:

$$\text{MRR} = \frac{1}{|Q|} \sum_{i=1}^{|Q|} \frac{1}{\text{rank}_i}, \text{NMR} = \frac{1}{|Q|} \sum_{i=1}^{|Q|} \frac{\text{rank}_i - 1}{N - 1}. \tag{24}$$

where $Q$ is the set of test queries, and $\text{rank}_i$ is the rank of the correct target class for the $i$-th test query. These metrics, in combination with accuracy, provide a more comprehensive evaluation framework for assessing conditional consistency evaluation metrics in generative models.

In this paper, we propose *Protein-MMD*, a new evaluation metrics that calculate the Maximum Mean Discrepancy (MMD) based on protein embeddings. Specifically, both real and generated protein sequences are encoded using the ESM2 language model (Lin et al., 2023), which provides biologically informed embeddings. In the embedding space, *Protein-MMD* captures both distributional and functional similarities between real and generated proteins more effectively, aligning with the goals of *de novo* protein design:

$$\text{Protein-MMD}(p_r, p_g) = \left\| \frac{1}{n} \sum_{i=1}^{n} \varphi(x_i) - \frac{1}{m} \sum_{j=1}^{m} \varphi(y_j) \right\|_{\mathcal{H}}^2, \tag{25}$$

where $\varphi(\cdot)$ denotes the embeddings extracted from the language model. These embeddings represent both sequence and functional information, making them particularly well-suited for comparing real and generated protein distributions.

To validate the effectiveness of Protein-MMD and other metrics, we apply the evaluation metrics on the Enzyme Commission (EC) dataset, which categorizes proteins based on the reactions they catalyze using EC numbers. We focus on seven classes from the first EC number category for our conditional generation

Table 1: Evaluation on the EC dataset.

| Metric | Accuracy ↑ | MRR ↑ | NMR ↓ |
|---|---|---|---|
| MMD | 0.0687 | 0.3101 | 0.5506 |
| Protein-FID | 0.2988 | 0.4825 | 0.3920 |
| **Protein-MMD** | **0.5487** | **0.6629** | **0.2524** |

task. In Table 1, we compare three evaluation metrics: MMD (considering only sequence statistics as presented in (Kucera et al., 2022)), Protein-FID (using ESM2 in place of Inception for protein generation), and Protein-MMD. All metrics are used to compute the Accuracy, Mean Reciprocal Rank (MRR), and Normalized Mean Rank (NMR) scores to evaluate how it performs in evaluating the conditional consistency. As observed, *Protein-MMD* outperforms both MMD and Protein-FID across all evaluation metrics. The higher accuracy and MRR scores indicate that *Protein-MMD* better captures the conditional consistency of the proteins in the test set. The lower NMR score further demonstrates that *Protein-MMD* ranks the correct target class higher in comparison to other metrics, validating its effectiveness in conditional protein generation tasks. While *Protein-MMD* proves to be the most effective metric according to our framework, we acknowledge the widespread use of FID in generative modeling tasks. Therefore, we will continue to report Protein-FID results alongside Protein-MMD in subsequent experiments.

## 4 EXPERIMENTS

### 4.1 EXPERIMENTAL SETUP

We compared our model against several baselines, each representing distinct approaches to protein generation. ProteoGAN (Kucera et al., 2022) is a GAN-based method, while ESM2 (Lin et al., 2023) and ProstT5 (Heinzinger et al., 2023) are Transformer-based language models specifically designed for protein sequence modeling. ProteinMPNN (Dauparas et al., 2022) and LatentDiff (Fu et al., 2024), on the other hand, are graph-based models, with LatentDiff also incorporating a diffusion-based framework, specifically using a latent diffusion approach. For each model, we evaluate the performance using both diversity metrics (TM-score, RMSD, and Seq.ID) and conditional consistency metrics (Protein-MMD and Protein-FID). Higher RMSD, lower TM-score, and lower Seq.ID indicate higher diversity, while lower Protein-MMD and Protein-FID values signify higher conditional consistency between the generated and real protein distributions.

To verify the effectiveness of our proposed multi-level conditional diffusion model, we conducted comprehensive experiments on two standard datasets: the Enzyme Commission (EC) dataset and the Gene Ontology (GO) dataset. The EC dataset categorizes proteins based on the biochemical reactions they catalyze, while the GO dataset classifies proteins according to their associated biological processes, cellular components, and molecular functions. These datasets provide a robust benchmark for assessing both the diversity and conditional consistency of generated protein sequences.

Our model leverages the `esm2_t33_650M_UR50D` model from ESM2 (Lin et al., 2023) as the amino acid-level encoder. To construct the Protein Variational Auto-encoder model, we set the latent dimension to 384, and the decoder is composed of 8 Transformer (Vaswani, 2017) decoder blocks, each equipped with an 8-head self-attention mechanism. The Protein Variational Auto-encoder model is trained with a learning rate of $10^{-4}$, using a combination of mean squared error (MSE) and cross-entropy as the loss functions. To regulate the latent vector distribution, we apply a KL divergence loss with a weight of $10^{-5}$. We experimented with 128, 256, and 512 as the maximum sequence lengths. For the diffusion model, we modify the DiT-B architecture from DiT (Peebles & Xie, 2023b), which consists of 12 DiT blocks and uses a hidden size of 768. The DiT model is trained from scratch with a learning rate of $10^{-4}$ and includes a weight decay of $10^{-5}$.

### 4.2 RESULTS AND ANALYSIS

Table 2 presents the results of our model and the baselines on two datasets. Our model achieves the best performance in terms of most metrics on the EC dataset, indicating superior conditional consistency and diversity in generating proteins that adhere closely to the specified enzyme classes.

On the EC dataset, our model (with sequence length 512) achieves the lowest Protein-MMD and Protein-FID scores, demonstrating effective modeling of the distributional and functional similarities between generated and real proteins. The RMSD and TM-score metrics indicate that our model generates structurally diverse proteins, with the highest RMSD and among the 2nd-lowest TM-scores, suggesting less topological similarity to templates. The sequence identity (Seq.ID) is also low, indicating higher sequence diversity.

For the GO dataset, our model also performs competitively. However, in terms of conditional consistency metrics (Protein-MMD and Protein-FID), our model ranks second, with ESM2 achieving the best Protein-MMD score and ProteoGAN achieving the best Protein-FID score. This suggests that while our model generates diverse protein structures on the GO dataset, it is slightly less effective in capturing the functional annotations compared to the top-performing models.

Table 2: Results on EC and GO datasets

| | EC Dataset | | | | | GO Dataset | | | | |
| | Diversity | | | Conditional Consistency | | Diversity | | | Conditional Consistency | |
| | TM-score↓ | RMSD↑ | Seq.ID↓ | Protein-MMD↓ | Protein-FID↓ | TM-score↓ | RMSD↑ | Seq.ID↓ | Protein-MMD↓ | Protein-FID↓ |
|---|---|---|---|---|---|---|---|---|---|---|
| ProteGAN | 0.26 | 5.35 | 6.71 | 13.99 | 260.31 | 0.23 | 5.96 | **6.33** | **10.89** | **256.31** |
| ESM2 | 0.29 | 4.25 | **6.57** | 13.35 | 238.46 | 0.22 | **7.33** | 6.39 | 11.86 | 290.31 |
| ProstT5 | 0.28 | 4.25 | 6.61 | 13.76 | 248.32 | 0.26 | 6.81 | 6.73 | 11.93 | 292.58 |
| ProteinMPNN | **0.24** | 4.24 | 67.43 | 22.31 | 587.72 | **0.14** | 7.10 | 77.96 | 15.94 | 410.43 |
| LatentDiff | 0.37 | 2.73 | 7.67 | 13.43 | 256.75 | 0.31 | 4.26 | 7.37 | 12.66 | 346.40 |
| Ours(128) | **0.24** | 4.7 | 7.56 | 13.74 | 250.2 | | —— | | | |
| Ours(256) | 0.27 | 4.40 | 6.88 | 13.67 | 248.31 | | —— | | | |
| Ours(512) | 0.25 | **5.39** | 6.79 | **13.28** | **237.46** | 0.26 | 6.09 | 7.13 | 11.67 | 284.65 |

The best performance for each metric is indicated in **bold**, while the second-best performance is underlined.

**Case Study.** To further demonstrate the superiority of our model on the GO dataset, particularly regarding conditional consistency, we conducted a fine-grained case study comparing our method with ProteoGAN. While our model leads on the EC dataset, it ranks second to ProteoGAN on the GO dataset in terms of conditional consistency metrics. We utilized an *in-silico* evaluation to perform a fine-grained analysis of the generated protein sequences. By employing a trained ESM-MLP classifier on the GO dataset, we assessed each generated protein's adherence to the specified GO terms using the Intersection over Union (IoU) (Rezatofighi et al., 2019; Gao et al., 2023b) metric.

As shown in Table 3, our method exhibits a lower average $\text{IoU}_{\text{mean}}$ compared to ProteoGAN, aligning with earlier results in Table 2. However, it achieves a higher $\text{IoU}_{\text{max}}$, indicating a greater potential for generating high-quality samples that closely match the desired GO annotations. Figure 4 illustrates the distribution of IoU scores. While ProteoGAN's samples are concentrated around medium quality, our method generates a broader range of samples, including those with higher IoU scores. This suggests that our model, despite a lower average performance, is more capable of producing proteins with superior conditional consistency.

Table 3: Comparison with ProteoGAN

| Method | $\text{IoU}_{\text{mean}}\uparrow$ | $\text{IoU}_{\text{max}}\uparrow$ |
|---|---|---|
| ProteoGAN | **0.2181** | 0.4706 |
| Ours (512) | 0.2088 | **0.5833** |

We also present visualizations of conditionally generated proteins by our method (Ours) and other baselines on the EC dataset, see Appendix A.2 for details.

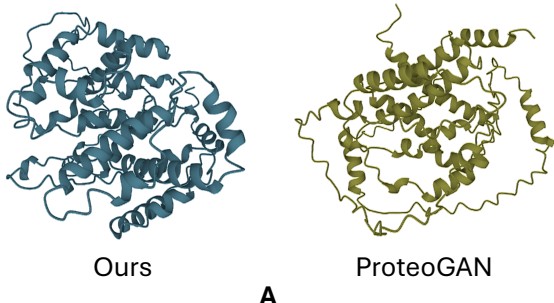
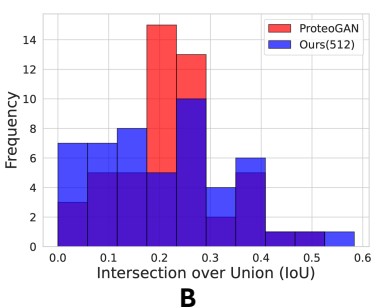

Figure 4: **A** shows the two highest generated protein results of Ours and ProteoGAN in terms of IoU indicator. **B** shows the statistical frequency histogram.

### 4.3 IMPACT OF MAXIMUM SEQUENCE LENGTH ON CONDITIONAL CONSISTENCY

In previous studies on protein *de novo* design, existing works usually employ a maximum sequence length of 128 (Fu et al., 2024). However, through our experiments, we observed that for conditional generation tasks, shorter sequence lengths fail to fully leverage the conditional information, which in turn results in lower conditional consistency metrics. To address this, we constructed models with three different maximum sequence lengths: 128, 256, and 512, and investigated the impact of maximum length on the model's ability to maintain conditional consistency.

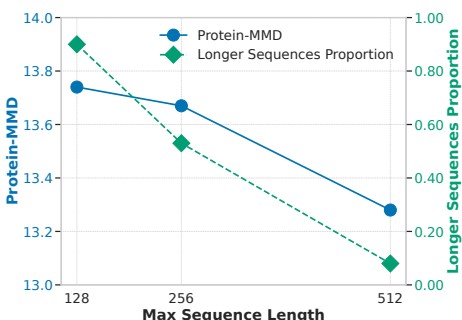

Figure 5: Effect of maximum length of sequence on Protein-MMD

As shown in Figure 5, we observe a positive correlation between the Protein-MMD metric, which reflects conditional consistency, and the proportion of training data samples exceeding the current maximum sequence length. This indicates that longer sequences help the model better incorporate condition information during generation. Moreover, the results in Table 2 for our method with different lengths reveal that the maximum sequence length does not influence the model's performance on diversity metrics, which are independent of the quality of condition-guided generation. These findings underscore the importance of maximum sequence

length in enhancing conditional consistency, offering valuable insights for the design of future protein conditional generation models.

### 4.4 ABLATION STUDY

To investigate the contribution of each of the three levels(amino acid, backbone, all-atom), we conducted an ablation study experiment. Specifically, the variant of our model removes either a specific level (the backbone or all-atom) or both two levels. Then we examine the

Table 4: Ablation Study

| Method | Protein-MMD↓ | Protein-FID↓ |
|---|---|---|
| All | **13.50** | **241.82** |
| Removed backbone level | 13.73 | 249.14 |
| Removed all-atom level | 13.94 | 251.83 |
| Removed both | 14.06 | 255.15 |

performance of the conditional consistency metrics. Note that we can not remove the amino acid level because the amino acid is required for evaluation. The ablation study is conducted on the EC dataset. As shown in Table 4, if we remove any level (i.e., backbone and all-atom level) or both two levels, the performance will drop. It verifies the necessity of our multi-level conditional diffusion.

## 5 RELATED WORKS

*De novo* protein design methods are dedicated to identify novel proteins with the desired structure and function properties (Watson et al., 2023; Huang et al., 2016; Frey et al., 2024; Mao et al., 2024; Komorowska et al., 2024; Gao et al., 2024). Recent advancements in machine learning have enabled a generative model to accelerate key steps in the discovery of novel molecular structures and drug design (Gao et al., 2023a; Lu et al., 2022; Pei et al., 2023; Liu et al., 2022). A prior step of generate models to the representation of proteins (Zhang et al., 2023; Zhou et al., 2023; Gong et al., 2024; Zhao et al., 2024; Liu et al., 2023; Somnath et al., 2021; Jamasb et al., 2024).

The majority of representation learning for protein is to represent a protein as a sequence of amino acids (Chen et al., 2023; Moreta et al., 2022; Lee et al., 2024; Fan et al., 2023). Considering the spatial information is important to the property of a protein, many works resort to a graph model for a comprehensive presentation with the structure information (Ingraham et al., 2019; Huang et al., 2024a). In general, each node on the graph is an amino acid and the edge is decided by the distance between two nodes Aykent & Xia (2022); Xia & Ku (2021); Hladis et al. (2023). Despite the power of graph models, the relation information in a 3-dimensional space captures the multi-level structure such as the angle between two edges. A line of research works explore the protein structure in 3D space Hermosilla et al. (2021); Huang et al. (2024b); Wang et al. (2023); Zhong et al. (2020); Peng et al. (2022); Liu et al. (2022); Huang et al. (2024a). Recently, large language models (LLMs) have also been introduced to model the sequence Meier et al. (2021); Lin et al. (2023); Hu et al. (2022); Su et al. (2024) inspired by the success of natural language processing.

Capitalizing on the power of generative models such as Generative Adversarial Networks (GANs) and diffusion models, deep generative modeling has shown its potential for fast generation of new and viable protein structures. Anand & Huang (2018) has applied GANs to the task of generating protein structures by encoding protein structures in terms of pairwise distances on the protein backbone. Diffusion models have emerged as a powerful tool for graph-structured diffusion processes Klarner et al. (2024). FrameDiff has been proposed for monomer backbone generation and it can generate designable monomers up to 500 amino acids Yim et al. (2023). NOS is another diffusion model that generates protein sequences with high likelihood by taking many alternating steps in the continuous latent space of the model Gruver et al. (2023).

## 6 CONCLUSIONS

In this paper, we introduce a novel multi-level conditional generative diffusion model that integrates sequence-based and structure-based information for efficient end-to-end protein design. Our model incorporates a 3D rotation-invariant preprocessing step to maintain SE(3)-invariance. To address the limitation of existing evaluation metrics, we propose a novel metric to evaluate conditional consistency for protein generation tasks by leveraging language models, which we hope to catalyze the research progress on protein generation.

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

# A APPENDIX

## A.1 PROTEIN-MMD

**Theorem:** $\exists N \in \mathbb{N}^+, \forall i > N, d(\mathbf{x}^i, C^k) < \min_{j \neq k} d(\mathbf{x}^i, C^j)$ where $\mathbf{C}^j$ is any other class.

*Proof*: Assume that there exists a class $C^j (j \neq k)$ such that $d(\mathbf{x}^i, C^j) \leq d(\mathbf{x}^i, C^i)$ for $i > N$. Since $C^k$ is defined as the correct target class and the quality of the generated sample improves with $i \to \infty$, the consistency $d(x_n^i, C^k)$ should approach zero. If $d(\mathbf{x}^i, C^j) \leq d(\mathbf{x}^i, C^i)$, we have $\lim_{n \to \infty} d(\mathbf{x}^i, C^j) = 0$. It contradicts the assumption that $\mathbf{C}^k$ is the correct class for the generated data. Therefore, the assumption is false.

## A.2 VISUALIZATION OF GENERATED RESULTS

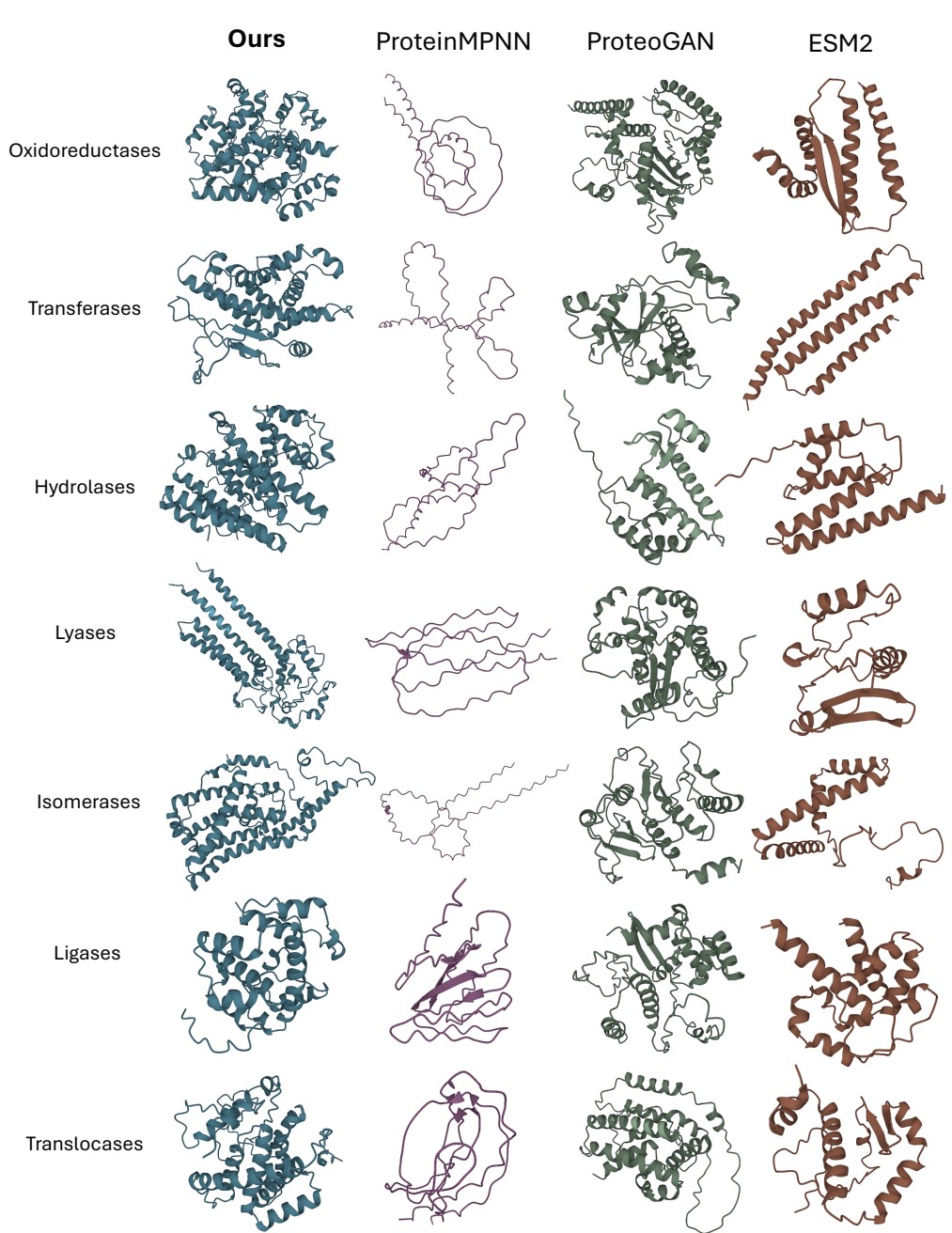

Figure 6: Protein Visualization Comparison on EC Dataset (Ours vs. ProteinMPNN, ProteoGAN, and ESM2)

