# OpenReview forum: "Towards Building Reliable Conditional Diffusion Models for Protein Generation"
_ICLR.cc/2025/Conference — ICLR 2025 Conference Withdrawn Submission_

### Official Review · Reviewer_6YnY · 2024-10-28

**Soundness:** 3
**Presentation:** 3
**Contribution:** 3
**Rating:** 6
**Confidence:** 4

**Summary:**

The paper introduces a multi-level conditional generative diffusion model for protein design, which incorporates both sequence-based and structure-based information. This approach aims to generate proteins with specific functional properties, addressing challenges in modeling the intricate relationships between protein structure and function. The model features a novel multi-level architecture that processes information at the amino acid, backbone, and atom levels, ensuring SE(3)-invariance through rotation-invariant preprocessing. Additionally, the authors propose Protein-MMD, a new evaluation metric based on Maximum Mean Discrepancy, to better assess the generated proteins' conditional consistency and functional relevance. Experimental results show that this framework enhances both diversity and consistency, outperforming existing models on standard datasets.

**Strengths:**

Originality: The multi-level approach and Protein-MMD metric bring innovation by tackling limitations in single-level generative models and evaluation, ensuring both structural and functional consistency.

Quality: The methodology is robust, featuring SE(3)-invariance and rigorous experiments across datasets, with ablation studies demonstrating the impact of each model component.

Clarity: Well-organized with clear diagrams and mathematical detail, though certain complex concepts could use brief explanations for broader accessibility.

Significance: This work is impactful for de novo protein design and applications like drug discovery and enzyme engineering, setting a foundation for improved protein generation and evaluation standards.

**Weaknesses:**

1. Interpretability of Protein-MMD and Model Outputs
The Protein-MMD metric is a novel contribution, yet its interpretability in biological terms could be enhanced. Currently, it primarily quantifies consistency without providing insight into which features drive functional adherence, making it challenging for practitioners to understand how generated proteins meet functional criteria.

2. Limited Scalability Testing on Complex Protein Types
The model is trained and evaluated primarily on relatively short sequences, leaving questions about scalability to more complex structures like large proteins or multi-domain proteins.

**Questions:**

1. Could you provide more details on how Protein-MMD captures functional adherence? Are there specific features or properties that most strongly influence this metric?
2. Have you tested the model on longer or more complex protein sequences (e.g., multi-domain proteins)? If so, what were the results?
3. How adaptable is the model to different protein language models, such as ProtT5 or other embeddings?
4. Could you elaborate on how information from each level (amino acid, backbone, atom) contributes to overall performance and functional adherence?
5. Could you demonstrate the practical benefits of SE(3)-invariance through experiments? How does it impact the model’s robustness and structural accuracy?
6. Are there specific real-world cases or benchmarks where you envision this model excelling, such as enzyme engineering or antibody design?

---

### Official Review · Reviewer_Aybd · 2024-10-31

**Soundness:** 2
**Presentation:** 1
**Contribution:** 2
**Rating:** 3
**Confidence:** 5

**Summary:**

This paper introduces an innovative multi-level conditional generative diffusion model for end-to-end protein design under specific functional guidance. The model integrates multi-level information from sequences and structures using the Diffusion-Transformer (DiT) architecture, combined with 3D rigid body rotational invariant preprocessing to ensure accurate structural modeling in three-dimensional space. By simultaneously generating representations at multiple levels, including amino acid, backbone, and full-atom levels, the model effectively captures hierarchical relationships within protein structures, providing more informative representations for the generated proteins.
In addition, a new evaluation metric named Protein-MMD is proposed, based on maximum mean discrepancy (MMD), to assess the conditional consistency of generated proteins. Experimental results demonstrate that Protein-MMD outperforms existing methods in evaluating distribution and functional similarity, providing a more reliable measure of the quality and conditional consistency of generated proteins.

**Strengths:**

This paper provides a new perspective on protein design by encoding proteins at multiple levels, including the amino acid level, backbone level, and atomic side-chain level. It learns feature distributions from the latent space to enable feature reconstruction and protein decoding during the reverse diffusion process.

**Weaknesses:**

1. There is a critical lack of description in the algorithm. For instance, while the authors claim to generate full-atom proteins, there is no detailed mathematical description of the encoding process at the atomic level.
2. The main text devotes extensive space to proving the SE(3) invariance of DiT, yet it neglects the more crucial property of equivariance in protein modeling. The authors instead align all proteins into a common global coordinate frame using an encoding method, which is clearly not an optimal approach.
3. The paper lacks any description of how conditional information is encoded and trained. For instance, in ESM3, considerable effort is dedicated to explaining how GO annotation information is encoded as tokens to conditionally control sequence generation, which itself is a challenging task.
4. There is no discussion of some of the challenges in the latent space diffusion of proteins, such as how to ensure that structures reconstructed from the latent space do not have clashes. The paper lacks discussions and analyses of these more pertinent issues.
5. The experimental setup raises doubts about the objective of the proposed diffusion model. The authors claim it to be "a novel multi-level conditional generative diffusion model that integrates sequence-based and structure-based information for protein design," and Figure 1 depicts a full-atom protein design process. However, the experimental task only focuses on generating sequences under given conditions (e.g., GO). There is no comparison with well-known sequence-structure diffusion methods in the field, such as Multiflow, for structure-related tasks.

**Questions:**

1. How are atomic-level features specifically encoded? The paper does not provide a detailed explanation.
2. How is a specific protein decoded from the latent space? Beyond sequence decoding, how is the structure, as shown in Figure 1, generated? This process lacks a clear description.
3. If we disregard the complexity of conditional information, can the model unconditionally generate a full-atom protein (including sequence, backbone, and side chains) from Gaussian noise in the latent space? If it can, what is the quality of the side-chain distribution? Do the generated sequence and structure match?
4. Disregarding the complexity of conditional information or selecting the best-performing condition, how does the model perform in terms of the traditional evaluation metrics for protein diffusion—designability, diversity, and novelty?
5. If the model is limited to sequence design, how does it compare to methods such as ProteinGenerator and ProtGPT2?

---

### Official Review · Reviewer_P7j5 · 2024-11-03

**Soundness:** 2
**Presentation:** 2
**Contribution:** 2
**Rating:** 5
**Confidence:** 4

**Summary:**

The authors propose a multi-level generative model which learns sequence, backbone and atomistic level information individually via graph representation where information is shared between levels. They claim that existing generative models are task specific, limiting adaptability and scalability across different protein families. To better asses the quality of conditionally generated proteins, they present protein-MMD, a maximum mean discrepancy metric applied to protein embeddings. They claim this metric outperforms existing metrics used to measure diversity and functional coverage.

**Strengths:**

- The proposal of a conditional flow mechanism of passing information from atom to sequence is very interesting.
- Their protein-MMD metric could be a good contribution to the current toolbox for evaluating generated protein sequences.

**Weaknesses:**

- The utility and execution of the multi-level conditioning model lacks detail and clarity. The authors propose this method specifically “to capture structural and functional nuances of protein sequences”, but do not provide sufficient evidence and ablations that this method would accomplish this superior to existing ones.
- The protein-MMD metric is missing important evaluations.

**Questions:**

1) Regarding the multi-level conditioning there are a number of important details missing from the methodology.
- What is $h_i$ (abstracts biochemical or structural properties), is this a learned or assigned value?
- At the backbone level, what is represented as edges?
- At the atom level, what atoms are being represented? Two methods are cited, but it’s not clear what is done in this paper. Not all PDBs have hydrogen bonds for example, so if those are included it must be clarified. Similarly, “the first four torsion angles” are not standard nomenclature, and should be defined; the two standard known torsion angles, phi and psi are backbone torsions.

2) It's unclear why proteinMMD metric is compared to FID, since they seem to be fundamentally different metrics. Something like F1 micro- or macro average scores seem to be better aligned with the task.
- Where are the two sets of sequences being evaluated in Table1, is it an equal split from the EC dataset?
- What is the MMD baseline, and how is that different from proteinMMD is (first row, table 1). What does it mean to only consider sequence statistics.
- How are accuracy MRR and MNR calculated for FID since FID itself is not a class-based metric.
- There should be additional baselines here to help with intuition surrounding the metric, 1) How the metric performs on true known sequences, 2) how the metric varies with other embeddings (e.g. ProtT5) , and 3) how it performs with random class assignment.
- Lastly, it would be helpful to understand where this metric is powerful/weak, what are possible ways to interpret the various values.

3) Typo in Table 2, I believe it should read ProtT5 not ProstT5 in the first column.

4) The motivation for the models included in Table 2 is unclear. Why were these models chosen, not all of these are generative models (e.g. ProteinMPNN, and ESM2). Why are SOTA generative models like RFDiffusion not included in the benchmark.  Similarly, ESM2 should be included separately as an ablation, since your model itself is using ESM2 embeddings?

5) For the ablation study  4.4 you could inverse folding generated structures to get amino acid sequences for comparison purposes.

6) Figure 6 examples appear to be helix-dominant, is there an impact on structural quality/diversity from how the learning task is defined?

---

### Official Review · Reviewer_GPV7 · 2024-11-03

**Soundness:** 2
**Presentation:** 2
**Contribution:** 1
**Rating:** 3
**Confidence:** 4

**Summary:**

In this paper, the authors present a multi-level conditional generative diffusion model that combines sequence- and structure-based information for protein design. Additionally, they propose a Protein-MMD metric to assess conditional consistency in protein generation, validated through experiments on standard datasets.

**Strengths:**

1. The paper presents a potential solution to a highly relevant problem.
2. Ablation studies is performed for the proposed architecture.
3. The paper is easy to read.

**Weaknesses:**

1. The idea of applying the MMD metric to protein domain using pLM embeddings is not novel, as multiple studies have already explored this, employing metrics like FID and MMD ([1],[2], etc)
2. There is no experiment demonstrating that $d(x^i,C^k)< \min d(x^i,C^j)$ for the EC dataset, nor an analysis of which distance function satisfies this condition. Additionally, it would be beneficial to clarify why ESM embeddings were chosen for calculating the MMD metric, as well as comparative analysis between different ESM versions or alternative protein language models to support this choice.
3. ESM2, ProtT5, and ProteinMPNN (inverse folding) are non-conditional generative models, and there is no explanation of any adaptations made to enable conditional generation. The lack of code further limits the ability to understand how, ESM2, ProtT5, and ProteinMPNN were modified for this purpose.
4. Diversity metrics (TM-score, RMSD and Seq.ID):
   - The calculation method for these metrics is not explained.
   - Metrics such as TM-score and RMSD are compared only within generated samples rather than against ground truth samples for each EC class. Without a defined scale, it’s unclear if these values are meaningful; in principle, random sample generation could achieve diversity, but the goal is to match the diversity within each class, so these standalone values lack context.
5. Protein quality metrics, such as pLDDT, perplexity, and other relevant indicators, are not included in the evaluation.
6. Lack of architectural specifics and learning information details

[1] https://openreview.net/forum?id=y2hFt8YJDw

[2] https://arxiv.org/abs/2403.03726

Overall, the pipeline is unclear, seems lack novelty to me, limited evaluation and there is no reproducibility.

**Questions:**

1. Why were ESM embeddings chosen for calculating the MMD metric, and which specific version of ESM was used?
2. Why was a molecular encoder chosen? Wouldn't it be more appropriate to use a protein-specific encoder?
3. Why is the ProteinMMD metric not squared like the classic MMD metric?
4. What is the statement "the quality of the generated sample improve with $ i \rightarrow \infty$ " based on?
5. The statement " a sequence of generated samples $ x^i | i = 0, 1, 2, \dots , \infty  $ ordered by increasing quality" does not specify the quality metric used.
6. Why the Protein Variational AutoEncoder is not specified in the model architecture figure (Figure 1)?"
7. Information about the GO dataset is missing: specifically, the hierarchy level of the selected classes, the number of examples used, and the data split.
8. The statement "existing works usually employ a maximum sequence length of 128" does not align with models like RFDiffusion, Progen (an autoregressive model), or other GPT-like models.
9. The statement "This indicates that longer sequences help the model better incorporate condition information during generation" should be aligned with the dataset's length distribution.
10.Could you please specify how the proof of SE(3)-invariance of the transformation is applied within the model architecture?

Minor comments:
1. A description of the MMD metric should be included in the preliminaries section.
2. typo in $d(x^i,C^j)<=d(x^i,C^i)$ should be $d(x^i,C^j)<=d(x^i,C^k)$

---

### Official Review · Reviewer_s5wA · 2024-11-04

**Soundness:** 2
**Presentation:** 2
**Contribution:** 1
**Rating:** 1
**Confidence:** 5

**Summary:**

The paper proposes a multi-level conditional diffusion model for protein generation that operates simultaneously at the amino acid, backbone, and all-atom levels. The work introduces a conditioning mechanism that allows information to pass between these levels during generation, supported by a SE(3)-invariant preprocessing step. The Authors also propose a metric called Protein-MMD for evaluation. However, there are fundamental issues with both the technical approach and empirical validation.

**Strengths:**

1. The task of protein generation guided by functional descriptions is highly relevant and timely in the current research landscape.
2. The hierarchical approach to protein generation is conceptually appealing and biologically motivated.

**Weaknesses:**

1. Lack of technical clarity and reproducibility. Critical implementation details are missing throughout the paper (source code could have helped but is not provided). Dataset preparation and splitting procedures are not described, making this work unreproducible. Training details and hyperparameters are incompletely specified.

2. Questionable novelty of the Protein-MMD evaluation metric. The Authors claim Protein-MMD as novel, ignoring substantial prior work using similar metrics. The justification for using MMD is poorly motivated and raises methodological concerns. It is not clear how exactly the experiment for validation of the effectiveness of Protein-MMD is set up.

3. Inappropriate baseline selection. Most baseline models (4 out of 5) are not conditional generation models (ESM2, ProstT5, ProteinMPNN, LatentDiff). The procedures to adapt the chosen baselines to the conditional generation task are not described. Baselines are pretrained on different datasets, making comparisons invalid. Selecting appropriate conditional generation baselines, e.g. ProGen, and adapting existing architectures for the same task would greatly enhance the work.

4. Poor choice of the evaluation metrics.  There are no quality metrics, which are necessary, as e.g. randomly generated proteins will be diverse. It is not clear, against what set of proteins TM-score and RMSD is calculated. The TM-score and RMSD values for two subsets of this set should be provided. Seq.ID is not defined. For MMD and FID a scale should be provided by measuring MMD and FID distances between random subsets of the reference dataset, otherwise, comparing values is senseless.

5. Dataset and overfitting concerns. Both datasets used in the work are called standard but no citation or source is provided. Using small datasets (presumably, EC ~15k, GO ~32k samples) with large models (3 transformer diffusion models, 12 blocks each, plus 3 transformer decoder models) suggests potential overfitting.  There is no discussion of measures taken to prevent or detect overfitting.

6. Limited evaluation scope. The Authors do not explore unconditional generation capability.  Missing comparison with relevant conditional generation baselines. There is no ablation study on conditioning mechanisms.

**Questions:**

1. The SE(3)-invariant preprocessing steps  include positioning the first amino acid into the origin, then rotating the whole structure, so that the second one aligns with x-axis, and then rotating the whole structure to put the third amino acid on the xy-plane. Please, explain the phrase "This process is iteratively applied to all amino acids in the protein chain" (line 194).

2. When training the model is fed with ground truth rather than previous prediction on each level. This can lead to a discrepancies between training and inference. Have you tried to train second and third level models on the predictions of the previous ones at least part of the time?

3. In the experimental setup section the "Protein Variational Auto-encoder model" is mentioned. How and why do you use it and how do you train it?

4. Exploring unconditional generation, which naturally arises from the architecture of the proposed model would greatly improve the work. Moreover, it is much easier to train the baselines and make the comparison, as there is a vast corpus of works in the unconditional setting.

---

### Note · Authors · 2024-11-13

I have read and agree with the venue's withdrawal policy on behalf of myself and my co-authors.